# ITEM: Improving Training and Evaluation of Message-Passing based GNNs for top-$k$ recommendation

**Yannis Karmim**
*yannis.karmim@cnam.fr*
*Conservatoire national des arts et métiers, CEDRIC, Paris, France*

**Elias Ramzi**
*elias.ramzi@valeo.com*
*Conservatoire national des arts et métiers, CEDRIC, Paris, France*

**Raphaël Fournier S'niehotta**
*fournier@cnam.fr*
*Conservatoire national des arts et métiers, CEDRIC, Paris, France*

**Nicolas Thome**
*nicolas.thome@isir.upmc.fr*
*Sorbonne Université, CNRS, ISIR, Paris, France*

**Reviewed on OpenReview:** *https://openreview.net/forum?id=9B6LM2uoEs*

## Abstract

Graph Neural Networks (GNNs), especially message-passing-based models, have become prominent in top-$k$ recommendation tasks, outperforming matrix factorization models due to their ability to efficiently aggregate information from a broader context. Although GNNs are evaluated with ranking-based metrics, *e.g.* NDCG@k and Recall@k, they remain largely trained with proxy losses, *e.g.* the BPR loss. In this work we explore the use of ranking loss functions to directly optimize the evaluation metrics, an area not extensively investigated in the GNN community for collaborative filtering. We take advantage of smooth approximations of the rank to facilitate end-to-end training of GNNs and propose a Personalized PageRank-based negative sampling strategy tailored for ranking loss functions. Moreover, we extend the evaluation of GNN models for top-$k$ recommendation tasks with an inductive user-centric protocol, providing a more accurate reflection of real-world applications. Our proposed method significantly outperforms the standard BPR loss and more advanced losses across four datasets and four recent GNN architectures while also exhibiting faster training. Demonstrating the potential of ranking loss functions in improving GNN training for collaborative filtering tasks.

## 1    Introduction

Recommender systems have become an essential component in many online applications, helping users discover relevant and personalized content amid the overwhelming abundance of information available on the internet. Collaborative filtering is one of the most popular and widely adopted techniques for building recommender systems, which operates by leveraging the past behavior of users and the relationships among items to generate recommendations.

Graph Neural Network architectures have emerged as powerful methods for learning and representing complex data structures, particularly those that exhibit non-Euclidean properties such as graphs (Veličković et al. (2017);

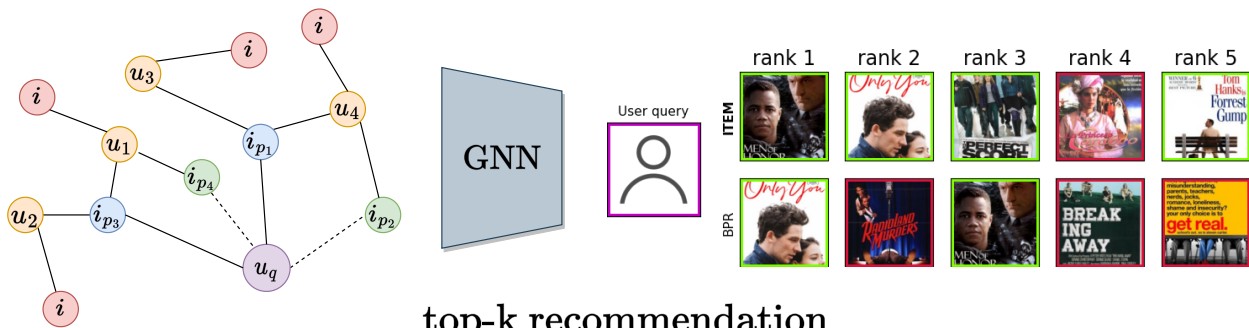

**top-k recommendation**

Figure 1: The goal in top-$k$ recommendation is to recommend to a user, *e.g.* $u_q$ (purple), relevant items such as $i_{p_4}$ (in green), based on its interaction history, *i.e.* items in blue such as $i_{p_1}$. ITEM directly optimizes the evaluation metric, *i.e.* NDCG, during training using a smooth approximation of the rank and Personalized PageRank (Page et al. (1998)) based negative sampling. Best seen in color.

Hamilton et al. (2017); Kipf & Welling (2016); Rossi et al. (2020); Xu et al. (2018)). In the context of recommender systems, GNNs offer a natural way to model user-item interactions, leveraging the inherent graph structure to capture higher-order relationships among users and items. Among the various GNN architectures, message-passing GNNs (MP-GNNs) have demonstrated impressive performance on top-$k$ recommendation tasks (He et al. (2020); Wang et al. (2019); Sun et al. (2020); Chen et al. (2020b)), consistently surpassing traditional matrix factorization methods (Rendle et al. (2012); Hsieh et al. (2017); Chen et al. (2020a)).

Top-$k$ recommendation is a standard task in recommender systems, which involves generating an ordered list of k items for a user. This process necessitates calculating similarities between elements before ranking them. Evaluation of top-$k$ recommendations typically employs ranking-based metrics such as Average Precision (AP), Normalized Discounted Cumulative Gain (NDCG), or Recall at k (R@k). However, these metrics are non-differentiable and cannot be directly used to train neural networks like GNNs with Stochastic Gradient Descent (SGD).

Instead, GNNs are commonly trained using a pairwise loss, the well-known Bayesian Personalized Ranking (BPR) (Rendle et al. (2012)), which serves as a coarse approximation of the ranking metric. The BPR is not explicitly designed to optimize standard non-differentiable evaluation rank-based metrics, such as NDCG@K and Recall@K: there are some criticisms about the gap between the evaluation objective and the training objective of the BPR (Wu et al. (2021)). We investigate alternative loss functions that more closely align the training and evaluation objectives.

The second issue arises from another misalignment, between batch learning and rank-based evaluation. During training, one does not have access to the true rank and must resort to hard negatives, that are more challenging to distinguish from positive items, for better performance. Some methods, such as MixGCF (Huang et al. (2021)), generate artificial hard negatives for the BPR loss. While this approach is suitable for a pairwise loss like BPR, its online process is computationally expensive. Given that we employ a listwise loss function, we need to sample a large number of informative negative items per user, the hard negative generation of MixGCF is not tractable. We thus propose to sample negative items offline, based on their Personalized PageRank (PPR) score (Page et al. (1998)): a high PPR score indicates proximity in the graph, making them more challenging to distinguish from a positive item for MP-GNNs.

Another impediment in the GNN literature for top-$k$ recommendation is the limited and unrealistic evaluation protocol. Recent works predominantly use a transductive approach, evaluating on the same users used in training. This not only deviates from real-world recommendation scenarios but also fails to consider GNN models' generalization capacity. We propose to enhance evaluation by incorporating an inductive user-split protocol, evaluating models on users not seen during training.

In this paper, we introduce our framework ITEM (**I**mproving **T**raining and **E**valuation of **M**essage-passing-based GNNs). Our framework is designed to provide training and evaluation for Graph Neural Networks tailored to the top-$k$ recommendation task.

- Specifically, our list-wise loss $\mathcal{L}_{\text{ITEM}}$ first leverages smooth rank approximations, which have recently been revisited in image retrieval and machine learning Bruch et al. (2019); Brown et al. (2020); Ramzi et al. (2021), leading to good approximations of the evaluation metrics, such as the NDCG or AP.

- Additionally, we enhance our loss $\mathcal{L}_{\text{ITEM}}$ by incorporating a negative sampling strategy tailored for rank approximation losses and leveraging the graph data structure. This strategy is based on the Personalized PageRank (Page et al. (1998)) (PPR) score. We show that this sampling is particularly well suited to our loss $\mathcal{L}_{\text{ITEM}}$, since it allows for a fast sampling of many informative negative items. This sampling helps to build large efficient batches to better approximate the true ranking.

- Finally, we propose to evaluate and benchmark GNNs for top-$k$ recommendation in an inductive user-split protocol. While it is known in the field of recommendation and used by some traditional models (Meng et al. (2020); Liang et al. (2018)), this setting has not been used to evaluate GNN architectures. This user-split protocol is more realistic because it introduces new users in testing, thus better evaluating the generalization capacity of recommender systems.

We carry out extensive experimental validations in both transductive and inductive settings. Our results highlight the benefits of ITEM over the standard training with BPR loss in terms of time and performance across multiple GNN architectures. Moreover, we demonstrate that ITEM outperforms more advanced state-of-the-art loss functions, showcasing its effectiveness.

## 2 Related Work

### 2.1 Graph Neural Networks for Collaborative Filtering

We focus on collaborative filtering (CF) models applied to data with implicit feedback, where only connections between users and items are considered, without incorporating other informations such as rating. A common variant of CF is the top-$k$ recommendation task, where the goal is to identify a small set of items that are most relevant to a user's interests. In this context, Graph Neural Networks have shown impressive results, indeed they are directly suitable to this task since user-item interactions can be modeled by bipartite graphs. Within the family of graph neural network models, message-passing-based (Gilmer et al. (2017)) methods have demonstrated superior performance compared to traditional CF models. These include matrix factorization (MF) (Rendle et al. (2012); Hsieh et al. (2017); Chen et al. (2020a)), auto-encoders (Liang et al. (2018)), and node embedding models that rely on random walks for generating representations (Perozzi et al. (2014); Grover & Leskovec (2016)). MP-GNNs for CF learn user and item representations by propagating and updating their embeddings through the bipartite graph. The BPR loss is employed to optimize the user-item representations by ensuring that a user's embedding is closer to a positive item than to a negative one. In this sense several models of MP-GNNs have been designed such as NGCF (Wang et al. (2019)), LR-GCF (Chen et al. (2020b)), or DGCF (Wang et al. (2020)).
He *et al.* introduce LightGCN (He et al. (2020)) as a simplified version of NGCF, achieved by eliminating weight matrices and non-linear activation layers. Although LightGCN is less expressive, it proves to be highly effective and more efficient, with a considerably simplified training process. Subsequent research has focused on enhancing the training of these models: SGL-ED (Wu et al. (2021)) proposes to combine the standard BPR loss with a self supervised loss, while MixGCF (Huang et al. (2021)) artificially generates hard negatives embeddings for negative sampling to replace the random negative sampling in the standard BPR loss.

### 2.2 Evaluation protocol for top-$k$ recommendation

All these GNN models competed with matrix factorization methods thus, their learning and evaluation setups are transductive.

However, other evaluations and training protocols for recommendation exist (Liang et al. (2018)). In (Meng et al. (2020)), authors show that the evaluations of GNNs-based models are limited compared to all those existing in the literature. Instead of splitting the data based on interactions, one may split the data by user, with some users in the training set, and new users for testing (see Fig. 3). This setup is closer to real applications, and allows to produce recommendation to new users without re-training the model. It is also more challenging, requiring to construct representations for a new user without learning.

We propose here to evaluate our ITEM model and benchmark MP-GNNs baselines in this realistic setting which was not used before for MP-GNNs, to the best of our knowledge.

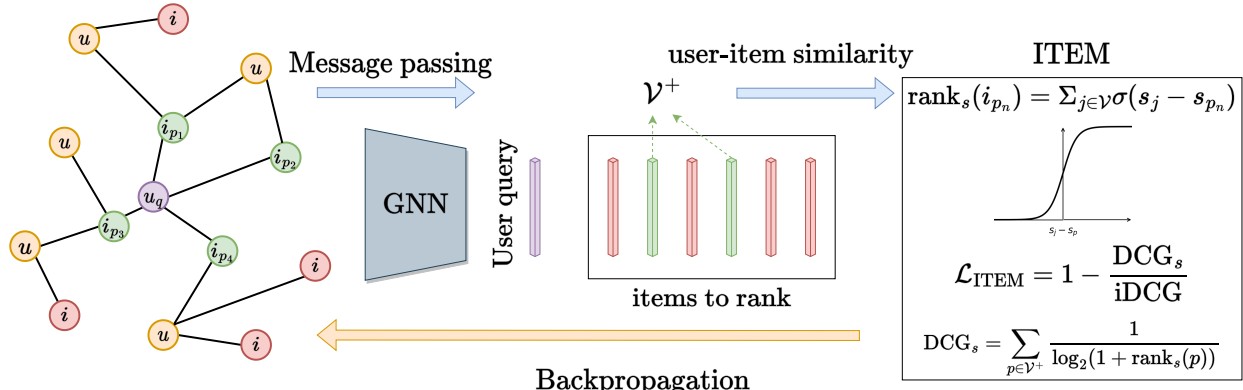

Figure 2: Using message passing, a GNN creates embeddings for every node of the graph. For each user we first construct a batch of randomly sampled positive items and negative items selected with our Personalized PageRank (Page et al. (1998)) based negative sampling Eq. (4). We then compute the score of the user wrt. the batch of items and calculate the loss using the approximation of the rank of Eq. (2). Finally the loss is backpropagated to update the parameters of the GNN, and update the embeddings for the items and users in the transductive setting. Best seen in color.

### 2.3 Ranking-based loss function

The Learning to Rank problem (Cao et al. (2007)) is evaluated using ranking based metrics, *e.g.* NDCG (Järvelin & Kekäläinen (2002)), R@K, or AP (Croft et al. (2010)). As these metrics are not differentiable (because of the ranking operator), their optimization has been abundantly studied. Different proxy methods have been built in Information Retrieval, with for instance pairwise loss (Burges et al. (2005); Hadsell et al. (2006)), or triplet losses (Rendle et al. (2012); Li et al. (2017)). To train GNNs, the most well-known loss is the BPR loss (Rendle et al. (2012)), a smoothed triplet loss. However, it was shown that triplet losses tend to put more emphasis on correcting errors at the bottom of the ranked list, rather than at the top, which would have the most impact to maximize the metric (Brown et al. (2020)). The *direct* optimization of ranking-based metrics has long been studied for Information Retrieval using structured SVM (Yue et al. (2007)), or rank approximations (Burges et al. (2006); Taylor et al. (2008); Qin et al. (2009)). Several direct optimization methods have gained traction in deep learning using for instance soft binning approaches (Revaud et al. (2019)) or rank approximations (Bruch et al. (2019); Pobrotyn & Białobrzeski (2021); Brown et al. (2020); Ramzi et al. (2021; 2022)). NeuralNDCG Pobrotyn & Białobrzeski (2021) proposed a differentiable approximation to NDCG by using NeuralSort, a differentiable relaxation of the sorting operator, resulting in a smooth variant of the metric and a new ranking loss function. ROADMAP Ramzi et al. (2021) introduce a robust and decomposable average precision (AP) loss for image retrieval, addressing non-differentiability and non-decomposability issues with a new rank approximation and calibration loss. Ramzi et al. (2022) use a hierarchical AP training method for pertinent image retrieval, leveraging a concept hierarchy to refine AP by integrating errors' importance and better evaluate rankings. In our work, we introduce the use of the sigmoid function for approximating the NDCG rank Brown et al. (2020); Qin et al. (2009) on GNNs in a top-$k$ recommendation context.

## 3 ITEM framework

In this section, we present the ITEM framework. We first define our ranking-based loss using a smooth approximation of the rank in Section 3.1 as well as our adapted negative sampling strategy in Section 3.2 . We then introduce the protocol used to train and evaluate the GNNs performances in Section 3.3.

**Training context** We consider an undirected bipartite graph $\mathcal{G} = (\mathcal{U}, \mathcal{V}, \mathcal{E})$ with $|\mathcal{U}|$ users, $|\mathcal{V}|$ items and $|\mathcal{E}|$ edges. We assign to each node in $\mathcal{U} \cup \mathcal{V}$ an embedding $\mathbf{h}$. We use a GNN that re-embeds the node embeddings to another space of the same dimension using message passing.

The task is to construct an embedding space such that, after message passing, the embedding of a user is closer to the embeddings of its positive items ($\mathcal{V}^+$) than to its negative items ($\mathcal{V}^-$), *i.e.* for user $u$, its embedding $\mathbf{h_u}$,

a positive item $p$, its embedding $\mathbf{h_p}$, and a negative item $j$, we want $s_p > s_j$, with $s_p = \mathbf{h_u} \cdot \mathbf{h_p}^T$, and $s_j = \mathbf{h_u} \cdot \mathbf{h_j}^T$, $s_p$ and $s_j$ are respectively the similarity score between the positive and the negative item. To evaluate the performances of a GNN, we use ranking-based metrics, NDCG Eq. (1), that measures the quality of a ranking.

$$\text{NDCG} = \frac{\text{DCG}}{\text{iDCG}}, \text{ with } \begin{cases} \text{DCG} = \sum_{p \in \mathcal{V}^+} \frac{1}{\log_2(1+\text{rank}(p))} \\ \text{iDCG} = \max_{\text{ranking}} \text{DCG} \end{cases} \tag{1}$$

### 3.1 Direct ranked-based optimization

GNNs are evaluated using standard ranking based metrics, *e.g.* NDCG, R@k, AP. We propose to train GNNs by optimizing directly smooth approximations of those metrics. Specifically, we use an approximation of the ranking operator to yield losses amenable to gradient descent.

The ranking operator can be defined as: $\text{rank}(p) = 1 + \sum_{j \in \mathcal{V}} H(s_j - s_p)$, with $H$ the Heaviside (step) function (Qin et al. (2009); Brown et al. (2020)). The intuition behind this definition is that in order to have the rank of a (positive) item $p$, we must "count" the number of items $j$ that have a similarity to the query user $s_j$ greater than $p$'s, $s_p$, *i.e.* $H(s_j - s_p) = 1$. During training, we aim to minimize $\text{rank}^- = 1 + \sum_{j \in \mathcal{V}^-} H(s_j - s_p)$, *i.e.* the number of "negative" items that have a higher score than positive items. Ranking-based metrics optimize this objective. This writing of the rank shows why the ranking operator is not differentiable, *i.e.* because the Heaviside function is not, specifically its gradients are null or undefined.

We propose to use the sigmoid function as an approximation of the Heaviside function (Qin et al. (2009); Brown et al. (2020)): $\sigma(x;\tau) = \frac{1}{1+\exp\frac{-x}{\tau}}$, with $\tau \in \mathbb{R}$ a temperature scaling parameter. $\tau$ controls the slop of the sigmoid, as $\tau$ gets smaller the slope is greater, and the sigmoid saturates faster.

Using this approximation we can define a smooth version of the rank:

$$\text{rank}_s(p,\tau) = 1 + \sum_{j \in \mathcal{V}} \sigma(s_j - s_p; \tau) \tag{2}$$

$\text{rank}_s$ is differentiable and is thus amenable to gradient descent. It has a single hyper-parameter, $\tau$, we study its impact in our experimental validation (Section 4.5).

**Application to ranked-based metrics** This plug-and-play rank approximation can be used to get a smooth version of ranking metrics, *e.g.* NDCG, R@k and AP.

The approximation of the Normalized Discounted Cumulative Gain (NDCG) is defined as follows:

$$\mathcal{L}_{\text{ITEM}} = 1 - \frac{\text{DCG}_s}{\text{iDCG}}, \text{ with } \text{DCG}_s = \sum_{p \in \mathcal{V}^+} \frac{1}{\log_2(1+\text{rank}_s(p))} \tag{3}$$

We use this approximation of $\text{rank}_s$ to approximate the DCG (see Eq. (1)), and use the exact iDCG (see Eq. (1)). In Sec. A of supplementary material we show how other ranking-based metrics can be approximated and show their effect on the evaluation metrics in Sec. B.5.

**End-to-end ranked-based training of GNN** GNNs, *e.g.* GCN (Kipf & Welling (2016)), jointly learn embeddings for the items and users of a graph, and through parameterized (or not for LightGCN He et al. (2020)) message passing they create representation for items and users. Using a user as a query (purple embedding in Fig. 2), as in Learning to Rank (Cao et al. (2007)), we aim to make the distance between a user's embeddings and the ones of its positive items, *i.e.* with implicit feedback (in green on Fig. 2), closer than the negatives ones, *i.e.* no implicit feedback (in red on Fig. 2). ITEM Eq. (3) is directly applied on the similarities, to produce better ranking. As ITEM is differentiable, after computing the loss we can backpropagate gradients through the network (see Fig. 2), to update the potential weights of the GNNs, and update the users and items embeddings.

### 3.2 Negative Sampling

The objective of ITEM is to rank, for a user query, its positive items (high rating or implicit feedback) before the negative ones. To do so, for each user we have to construct, a batch of positive items and negative items

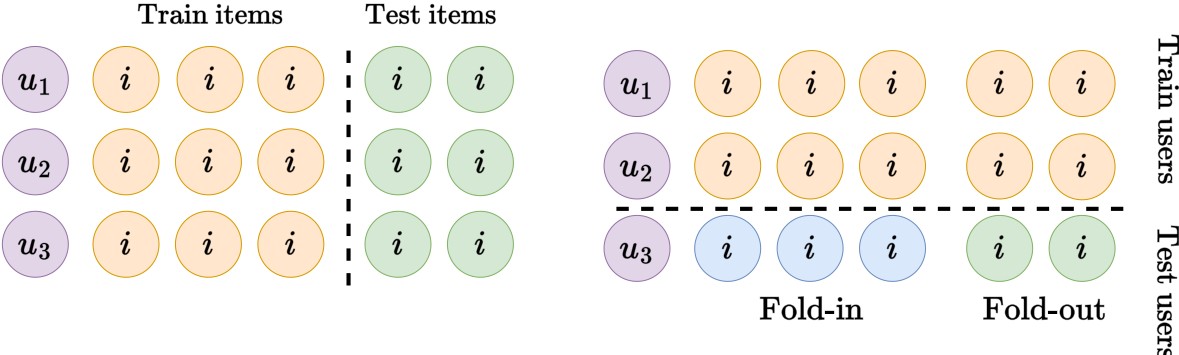

Figure 3: Transductive interaction-split (left) vs Inductive user-split (right) (Meng et al. (2020)). In the first case, the same users are in train and test, their learned embeddings can be directly used in test. In the second case, a part of the users and 100% of their interactions are used in train. During the evaluation, the model infers a representation of a new test user from some interactions (fold-in), in order to predict the fold-out items where we apply the ranking metrics.

as we cannot use all items. For a given user the number of negative items is much larger than that of the positive items. In order to better approximate the global ranking in mini-batch learning, we have to sample a significant number of informative negative items. Uniform random sampling is adopted as a solution in many GNN models for recommendation (He et al. (2020); Wang et al. (2019)), however the sampled negatives are often not very informative which can limit the performances of the model. We propose to use the Personalized PageRank (Page et al. (1998)) (PPR) to weigh the sampling of negative examples. Specifically we normalize the PPR score using the softmax function and sample negative items $j$ for a user query $u_q$ according to Eq. (4). Indeed items with high PPR score will be harder to rank, as they are in closer proximity to the user query.

$$p(i_j^- | u_q) \quad \sim \quad \frac{e^{\mathrm{ppr}_{u_q}(i_j^-)}}{\sum_{k \in \mathcal{V}^-} e^{\mathrm{ppr}_{u_q}(i_k^-)}} \tag{4}$$

Unlike PinSage (Ying et al. (2018)), our sampling strategy evaluates all negative items by weighing them differently. Specifically, the PPR score is calculated for each target user, assessing its proximity to every item in the bipartite graph. These scores are normalized with the softmax function to create a probability distribution for sampling. In contrast to existing sampling strategies such as MixGCF (Huang et al. (2021)) which generate hard negatives from embeddings "online" (i.e., during training), our PPR score can be computed on the bipartite graph "offline", before training. The offline computation makes hard negative sampling much faster, allowing us to build informative and large batches well-suited for our $\mathcal{L}_{\mathrm{ITEM}}$ loss. We perform ablative experiments in Table 5 to show the benefit of this sampling strategy. Our strategy efficiently selects informative hard negative items, but runs the risk of including false negatives. Alternative strategies may avoid false negatives entirely, but result in non-informative samples. Our results in Table 5 demonstrate that our trade-off, which samples hard negatives, in spite of the risk of false negatives, improves model performances.

### 3.3 Protocol for collaborative filtering

#### 3.3.1 Inductive protocol

We propose to evaluate GNNs in the top-$k$ recommending task using an inductive user-split protocol (Meng et al. (2020); Liang et al. (2018)) (illustrated on the right side of Fig. 3). During training, we have access to a fraction $\mu$ (i.e., $u_1$, $u_2$ on Fig. 3) of the user and their *entire* click history, for evaluation we consider the $1-\mu$ (i.e. $u_3$ on Fig. 3) unseen users and, with access to a fraction $\eta$ of their click history (*fold-in* – in blue on Fig. 3), we must recommend the other $1-\eta$ fraction (*fold-out* – in green on Fig. 3) of their click history. Despite the benefits of this inductive protocol known in top-$k$ recommendation, to our knowledge no GNN models are measured in this

Table 1: Statistics of the four datasets used for our experimental validation.

| Dataset | #Users | #Items | #Links |
|---|---|---|---|
| MovieLens-100K (Harper & Konstan (2015)) | 610 | 8957 | 100k |
| MovieLens-1M (Harper & Konstan (2015)) | 6022 | 3043 | 1m |
| Yelp-2018 (Asghar (2016)) | 31668 | 38048 | 1.5m |
| Amazon-book (Wang et al. (2019)) | 52643 | 91599 | 2.9m |

user-split protocol in this task. This protocol measures the capacity of a GNN to make accurate recommendation for a new user (not seen during training), based on its first interactions.

### 3.3.2 Transductive protocol

The standard protocol for evaluating GNNs in a recommendation task (Wang et al. (2019); He et al. (2020); Wu et al. (2021)) is to evaluate them in a transductive manner, *i.e.* items *and* users are the same during training and evaluation (left of Fig. 3). The training and evaluation splits are defined using an interaction-split, specifically for a given user we use a fraction $\rho$ (orange on Fig. 3) of its interaction to perform message passing during training and for evaluation the GNN must give a higher score to the $1-\rho$ (green on Fig. 3) fraction of its other interactions before the items to which it has no interaction with. For this protocol, the nodes in the graph remain the same for training and evaluation: the system cannot accommodate new users, it is limited to presenting new content to existing users.

## 4 Experiments

In our experiments and ablation studies, our aim is to demonstrate the effectiveness of our ranking loss function as well as our negative sampling setup applied to MP-GNNs in comparison to the standard BPR loss and other training enhancement strategies such as self-supervised regularization loss or alternative negative sampling techniques. Additionally, we propose to refine the evaluation of GNNs for CF by benchmarking them using an inductive, user-centric protocol.

### 4.1 Datasets and evaluation protocol

#### 4.1.1 Evaluation metrics

We follow (Huang et al. (2021); He et al. (2020); Wang et al. (2019)) evaluation protocol, computing the NDCG@20 and Recall@20 with the all-ranking protocol (He et al. (2017)).

#### 4.1.2 Datasets

We validate our method on four recommendation datasets. We use the MovieLens-100k and MovieLens-1M (Harper & Konstan (2015)) dataset, the 2018 edition of the dataset from the Yelp challenge (Asghar (2016)) and Amazon-book (Wang et al. (2019)). We detail the preprocessing of the datasets in Sec. B.1 in supplementary.

#### 4.1.3 Data splitting strategies

**Transductive setting:** For the transductive setup we use the exact same data preprocessing and split as the LightGCN (He et al. (2020)) and MixGCF (Huang et al. (2021)) models. In the transductive split, for each user 80% of these interactions have been randomly selected for the training set and the 20% for the test and validation set. We followed this same ratio across all datasets in our experiments. This data split is illustrated on the left side of Fig. 3.

**Inductive setting:** The inductive setup is based on the user-split data protocol, detailed in Meng et al. (2020) and used by the variational encoder model Mult-VAE (Liang et al. (2018)). We used the same ratio as in the Mult-VAE paper, *i.e.* In this protocol, we first separate users into training,validation and test sets. 80% of the users are kept for training, 10% for validation and 10% for testing. Unlike the test and validation sets, the training users keep 100% of their interaction history for learning our models. The test and validation user interactions

are separated into a *fold-in* and a *fold-out* set. The *fold-in* set is used to build a representation of the user in inference from a partial history, predictions and evaluation are performed on the *fold-out* set. In our experiments, the fold-in is composed of 80% of the randomly sampled interactions of a user, the rest is for the fold-out. This data split is illustrated on the right side of Fig. 3.

### 4.2 Baselines and implementation details

#### 4.2.1 Baselines

We use several GNN architectures GCN (Kipf & Welling (2016)), GAT (Veličković et al. (2017)), GIN (Xu et al. (2018)) and LightGCN (He et al. (2020)) and compare their original training, *i.e.* with the BPR (Rendle et al. (2012)), to ITEM training. To assess our model against the state of the art, we compare it with various CF model families, such as MF (MF-BPR (Rendle et al. (2012)), ENMF (Chen et al. (2020a))), metric-learning based (CML (Hsieh et al. (2017))), graph embedding (DeepWalk (Perozzi et al. (2014)), LINE (Tang et al. (2015)), Node2Vec (Grover & Leskovec (2016))), VAE models (Mult-VAE (Liang et al. (2018))), MP-GNNs (NGCF (Wang et al. (2019)), DGCF (Wang et al. (2020)), LightGCN (He et al. (2020)), NIA-GCN (Sun et al. (2020)), LR-GCCF (Chen et al. (2020b))). We also compare ITEM to a recent embedding enhancement method Chen et al. (2023). Finally, we compare ITEM to other losses and negative sampling methods such as NeuralNDCG (Pobrotyn & Białobrzeski (2021), another ranking loss); SGL-ED (Wu et al. (2021)) which combines BPR and a self-supervised loss; XSimGCL and SimGCL (Yu et al. (2022)) two contrastive-based models for CF; MixGCF (Huang et al. (2021)) which employs a hard negative sampling method for BPR. For a fair comparison of the various loss functions, we employ the same LightGCN backbone across all models.

#### 4.2.2 Implementation details

We give all details on hyper-parameters and optimization procedure in Sec. B.2 of supplementary. For the inductive setting, we found experimentally that learning user embeddings during training is harmful for the generalization of the GNNs to new users, so for training and evaluation users' embeddings are inferred using message passing only.

### 4.3 Main results

In this section we present our main results, we compare ITEM vs state-of-the-art methods. We first compare ITEM on the standard transductive protocol in Table 3, we then compare ITEM on the indcutive protocol in Table 2. For both protocols we use a LightGCN (He et al. (2020)) backbone following (Huang et al. (2021); Wu et al. (2021)).

#### 4.3.1 Transductive state of the art comparison

In Table 3 we compare, in the transductive setting, ITEM using a LightGCN (He et al. (2020)) backbone vs state-of-the-art methods. We show that across all datasets ITEM outperforms all the MP-GNNs state-of-the-art methods NGCF, LR-GCCF, NIA-GCCN, LightGCN and DGCF. It surpasses non-graph methods on the three datasets with relative improvements of +18% NDCG@20 on Yelp-2018 or 21% NDCG@20 on Amazon-book vs Mult-VAE, which shows the interest of using dedicated architectures for graph learning. Furthermore, ITEM outperforms the recent NeuralNDCG (Pobrotyn & Białobrzeski (2021)), *e.g.* +0.89 R@20 and +0.65 NDCG@20 on Yelp-2018, as well as SGL-ED, *e.g.* +0.55 R@20 and +0.23 NDCG@20 on Amazon-Book, and the advanced sampling method MixGCF (Huang et al. (2021)), *e.g.* +2.45 R@20 and +2.64 NDCG@20 on MovieLens-1M. Despite the limitations of MP-GNNs (Alon & Yahav (2020); Chen et al. (2019)), ITEM shows competitive results against the hybrid model SimpleX (Mao et al. (2021)) with better results on two of three datasets with +1.78 R@20 on MovieLens-1M and +0.38 R@20 on Yelp-2018 with the best state of the arts results.

#### 4.3.2 Inductive state-of-the-art comparison:

MP-GNNs can be easily evaluated in an inductive setup, unlike matrix factorization models or random walk embedding methods. MP-GNNs have a strong ability to generalize to users not seen during the training phase with the propagation and update process. We show that our ITEM method significantly boosts the performance of MP-GNNs in the inductive setup. We compare ITEM using LightGCN (He et al. (2020)) to state-of-the-art

Table 2: Comparison of ITEM vs state-of-the-art methods on three ***inductive*** benchmarks. Best results in **bold**.

| Method | MovieLens-100K | | Yelp-2018 | | Amazon-Book | |
|---|---|---|---|---|---|---|
| | R@20 | N@20 | R@20 | N@20 | R@20 | N@20 |
| Mult-VAE (Liang et al. (2018)) | 30.14 | 28.28 | 10.15 | 8.18 | 10.86 | 9.2 |
| GCN (Kipf & Welling (2016)) | 28.74 | 27.68 | 7.34 | 5.76 | 8.85 | 7.61 |
| GAT (Veličković et al. (2017)) | 31.01 | 28.92 | 9.04 | 7.32 | 9.88 | 8.17 |
| GIN (Xu et al. (2018)) | 29.71 | 27.58 | 7.34 | 5.76 | 9.62 | 8.05 |
| LightGCN (He et al. (2020)) | 30.79 | 29.73 | 7.88 | 6.34 | 9.56 | 8.02 |
| NeuralNDCG (Pobrotyn & Białobrzeski (2021)) | 31.12 | 30.07 | 9.14 | 7.62 | 9.54 | 8.19 |
| MixGCF (Huang et al. (2021)) | 32.07 | 30.62 | 9.85 | 8.21 | 10.11 | 9.63 |
| **ITEM (ours)** | **33.84** | **32.63** | **10.54** | **8.70** | **11.03** | **9.89** |

Table 3: Comparison of ITEM vs state-of-the-art methods on three ***transductive*** benchmarks. Best results in **bold**.

| Method | MovieLens-1M | | Yelp-2018 | | Amazon-Book | |
|---|---|---|---|---|---|---|
| | R@20 | N@20 | R@20 | N@20 | R@20 | N@20 |
| MF-BPR (Rendle et al. (2012)) | 21.53 | 21.75 | 5.49 | 4.45 | 3.38 | 2.61 |
| CML (Hsieh et al. (2017)) | 17.30 | 15.63 | 6.22 | 5.36 | 5.22 | 4.28 |
| ENMF (Chen et al. (2020a)) | 23.15 | 20.69 | 6.24 | 5.15 | 3.59 | 2.81 |
| DeepWalk (Perozzi et al. (2014)) | 13.48 | 10.57 | 4.76 | 3.78 | 3.46 | 2.64 |
| LINE (Tang et al. (2015)) | 23.36 | 22.26 | 5.49 | 4.46 | 4.10 | 3.18 |
| Node2Vec (Grover & Leskovec (2016)) | 14.75 | 11.86 | 4.52 | 3.60 | 4.02 | 3.09 |
| Mult-VAE (Liang et al. (2018)) | 29.23 | 23.84 | 6.41 | 4.97 | 4.46 | 3.33 |
| NGCF (Wang et al. (2019)) | 25.13 | 25.11 | 5.79 | 4.77 | 3.44 | 2.63 |
| LR-GCCF (Chen et al. (2020b)) | 22.31 | 21.24 | 5.61 | 3.43 | 3.35 | 2.65 |
| NIA-GCN (Sun et al. (2020)) | 23.59 | 22.42 | 5.99 | 4.91 | 3.69 | 2.87 |
| LightGCN (He et al. (2020)) | 25.76 | 24.27 | 6.28 | 5.15 | 4.23 | 3.17 |
| DGCF (Wang et al. (2020)) | 26.40 | 25.04 | 6.54 | 5.34 | 4.22 | 3.24 |
| SGL-ED (Wu et al. (2021)) | 26.34 | 24.87 | 6.75 | 5.55 | 4.78 | 3.79 |
| SimGCL (Yu et al. (2022)) | 27.55 | 25.01 | 7.21 | 6.01 | 4.90 | 3.98 |
| XSimGCL (Yu et al. (2022)) | 27.94 | 25.36 | 7.33 | 6.06 | 5.02 | 4.11 |
| Adap-$\tau$ Chen et al. (2023) | 27.87 | 26.15 | 7.33 | **6.12** | **6.12** | **4.90** |
| NeuralNDCG (Pobrotyn & Białobrzeski (2021)) | 29.45 | 25.13 | 6.50 | 5.23 | 4.38 | 3.27 |
| MixGCF (Huang et al. (2021)) | 27.35 | 24.56 | 7.17 | 5.84 | 4.51 | 3.41 |
| **ITEM (ours)** | **29.80** | **27.20** | **7.39** | 5.88 | 5.23 | 4.02 |

methods, and show that on the three datasets, ITEM sets new state-of-the-art performances for inductive recommendation. It outperforms Mult-VAE (Liang et al. (2018)) which is designed for the inductive setting, by +3.79 NDCG@20 on MovieLens-100k, +1.4 NDCG@20 on Yelp-2018 and +0.69 NDCG@20 on Amazon-book. We compare ITEM to different GNNs that were optimized using the BPR loss, this is further studied in Tab. 1 of supplementary where we show the interest of the well designed loss of ITEM vs the standard BPR loss. Finally ITEM outperforms MixGCF (Huang et al. (2021)) on the three datasets, *e.g.* on MovieLens-100k with +1.77 R@20 or +1.58 R@20 on Yelp-2018 as well as the ranking loss function NeuralNDCG (Pobrotyn & Białobrzeski (2021)), *e.g.* +1.54 R@20 and +1.08 NDCG@20. Although message-passing models face certain challenges (Alon & Yahav (2020); Chen et al. (2019)), when evaluated with the proposed inductive protocol they outperform state-of-the-art methods and exhibit fast training and strong generalization capabilities for unseen users during the training phase.

### 4.4 Ablation studies

In this section we perform ablation studies for the two elements of ITEM. We first study the impact of using $\mathcal{L}_{\text{ITEM}}$ vs the BPR loss in Table 4. We then study the impact of the sampling in Table 5.

#### 4.4.1 Ranking loss vs BPR

In Table 4 we compare in the same settings the BPR loss (Rendle et al. (2012)) and our proposed ranking-based loss $\mathcal{L}_{\text{ITEM}}$ on three *transductive* benchmarks, and show that for all four considered architectures, ITEM outperforms the BPR loss. Using our loss in combination with LightGCN (He et al. (2020)), the state-of-the-art GNN for transductive datasets, increases relative performances from +10.05% NDCG@20 on MovieLens up to +22.1% NDCG@20 on Amazon-book. We can also note that when using ITEM more expressive GNNs, *e.g.* GAT (Veličković et al. (2017)), perform better than LightGCN on MovieLens-1M and Amazon-book. With GAT the relative improvements range from +18.55% NDCG@20 on MovieLens-1M up to +54.9% NDCG@20 on Amazon-book. Overall, Table 4 shows the huge benefits of optimizing a ranking-based loss rather than a *proxy* loss. Table 1 of supplementary shows similar results for the inductive benchmarks. In Table 1 of supplementary we show that the best performing architectures on the inductive setting are not the same as in the transductive setting.

Table 4: Comparison of our ranking-based loss ITEM vs the BPR loss (Rendle et al. (2012)), using different GNN architectures on 3 ***transductive*** benchmarks. For each architecture best results is **bold**, best overall results is underlined.

| Method | | MovieLens-1M | | Yelp-2018 | | Amazon-book | |
|---|---|---|---|---|---|---|---|
| | | R@20 | NDCG@20 | R@20 | NDCG@20 | R@20 | NDCG@20 |
| GCN | BPR | 27.26 | 22.75 | 5.21 | 4.04 | 3.48 | 2.55 |
| | $\mathcal{L}_{\text{ITEM}}$ | **29.96** | **25.62** | **6.81** | **5.33** | **4.89** | **3.70** |
| | %Improv. | +9.90% | +12.62% | +30.7% | +31.9% | +40.5% | +45.1% |
| GIN | BPR | 26.99 | 22.28 | 5.52 | 4.32 | 3.42 | 2.54 |
| | $\mathcal{L}_{\text{ITEM}}$ | **29.94** | **25.60** | **6.87** | **5.38** | **5.24** | **3.94** |
| | %Improv. | +10.93% | +14.90% | +24.5% | +24.5% | +53.2% | +55.1% |
| GAT | BPR | 27.50 | 22.96 | 5.08 | 3.89 | 3.41 | 2.53 |
| | $\mathcal{L}_{\text{ITEM}}$ | **30.43** | **27.22** | **7.03** | **5.47** | **5.20** | **3.92** |
| | %Improv. | +10.65% | +18.55% | +38.4% | +40.6% | +52.5% | +54.9% |
| LGCN | BPR | 25.76 | 24.27 | 6.26 | 5.14 | 4.23 | 3.17 |
| | $\mathcal{L}_{\text{ITEM}}$ | **29.69** | **26.71** | **7.25** | **5.70** | **5.09** | **3.87** |
| | %Improv. | +15.27% | +10.05% | +15.81% | +10.89% | +20.3% | +22.1% |

#### 4.4.2 Negative sampling

We show in Table 5 ablation studies for the impact of our PPR sampling strategy. On both datasets our sampling strategy boosts the performances of ITEM, *e.g.* +3.2% NDCG@20 relative improvement on Yelp-2018 and +3.9% NDCG@20 on Amazon-book. We also use the MixGCF (Huang et al. (2021)) negative sampling with ITEM, however due to its high computation overhead, we cannot sample many negatives. This leads to MixGCF negatively affecting performances on Yelp-2018 and Amazon-book, *e.g.* -0.34 NDCG@20 on Amazon-book.

Table 5: Ablation study of the components of ITEM. The loss: ✓ for $\mathcal{L}_{\text{ITEM}}$, BPR otherwise (✗). And sampling: random negative sampling vs MixGCF (Huang et al. (2021)) vs the ITEM sampling Eq. (4) (PPR).With LightGCN (He et al. (2020)) on ***transductive*** benchmarks.

| $\mathcal{L}_{\text{ITEM}}$ | MixGCF | PPR (ours) | Yelp-2018 | | Amazon-book | |
|:---:|:---:|:---:|:---:|:---:|:---:|:---:|
| | | | R@20 | NDCG@20 | R@20 | NDCG@20 |
| ✗ | ✗ | ✗ | 6.26 | 5.14 | 4.23 | 3.17 |
| ✓ | ✗ | ✗ | 7.25 | 5.70 | 5.09 | 3.87 |
| ✓ | ✓ | ✗ | 7.21 | 5.71 | 4.86 | 3.53 |
| ✓ | ✗ | ✓ | **7.39** | **5.88** | **5.23** | **4.02** |

### 4.5 Model analysis

#### 4.5.1 Efficiency comparison

In Table 6 we compare the training times of LightGCN (He et al. (2020)) using the BPR loss, MixGCF and ITEM on the Yelp-2018 dataset until convergence. We train these 3 variants with a Quadro RTX 5000 GPU. LightGCN-BPR takes 10h26 before convergence on the Yelp-2018 dataset, which contains 1.5 million interactions. With MixGCF, the convergence time is reduced to approximately 6 hours, as hard negative sampling accelerates convergence, despite the time per epoch being more than twice as long due to the online generation of the hard negatives examples.

Table 6: Computational cost on Yelp-2018 with LightGCN backbone.

| Method | Time / epoch | Training Time |
|:---|:---:|:---:|
| BPR | 53s | 10h26 |
| MixGCF | 145s | 6h07 |
| ITEM (ours) | **12.3s** | **1h43** |

Finally, ITEM leads to model convergence in just 1 hour and 43 minutes. ITEM performs far fewer iterations per epoch by building batches of negative and positive items per user to approximate the NDCG metric.

#### 4.5.2 Impact of $\tau$

We show in Figs. 4a and 4b the impact of $\tau$ in Eq. (2) on the R@20 and NDCG@20 when optimizing ITEM on MovieLens for the inductive protocol. We can see on both figures that our method is robust for a wide range of $\tau$. Specifically, for values of $\tau$ in range 0.1 to 2.0, ITEM outperforms the BPR loss. Also note that using a finer selection of $\tau$ could lead to better results than reported in Table 2, *e.g.* using $\tau = 0.3$ leads to a R@20 of 33.52 in Fig. 4a against 33.13 for the value of $\tau$ used in Table 2 (note that in Table 2 ITEM additionally uses the PPR sampling).

### 4.6 Qualitative results

In the qualitative results presented in Fig. 5, a comparative analysis between our approach and the BPR loss baseline reveals the superior ranking achieved by ITEM. This enhanced ranking is not solely attributable to optimizing the target metric directly during training; it is also a result of the efficacy of our hard negative sampling strategy. This strategy contributes to a more effective differentiation between negative and positive instances, further refining the ranking quality. Supplementary B.6 includes a detailed listing of the top-20 results for comprehensive examination.

## 5 Conclusion

In this study, we introduce the ITEM framework to optimize GNNs for the top-$k$ recommendation task. By incorporating ranking losses used in image retrieval and directly optimizing item ranking for a given user, we demonstrate the limitations of the BPR loss, a widely-used pairwise loss within the GNN community.

Our proposed list-wise loss requires sampling a substantial number of hard *negative* items for each user, which is a challenging task. Unable to employ *online* methods such as MixGCF (Huang et al. (2021)), we devised an *offline*

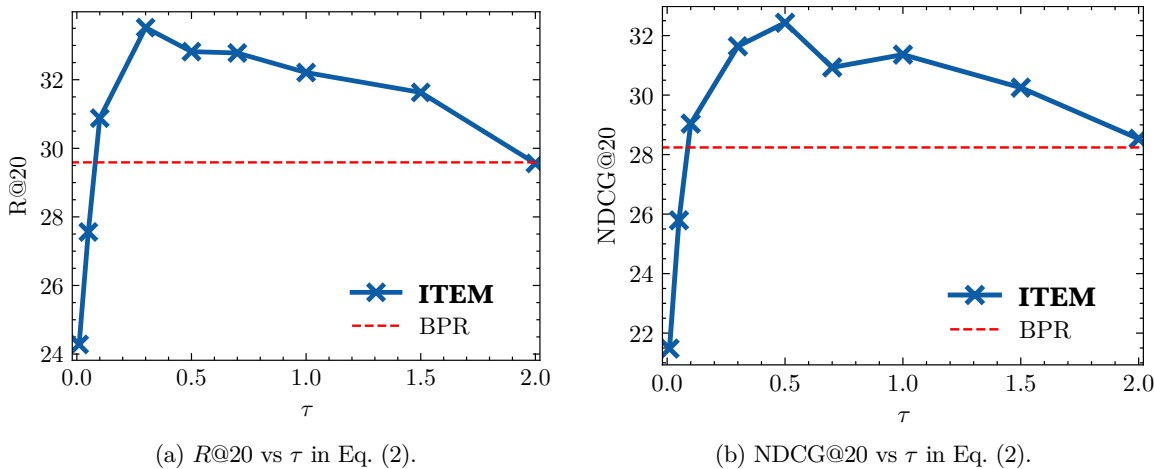

(a) $R$@20 vs $\tau$ in Eq. (2).  (b) NDCG@20 vs $\tau$ in Eq. (2).

Figure 4: $\tau$ in Eq. (2) vs R@20, NDCG@20 on MovieLens-100k (inductive) with LightGCN (He et al. (2020)).

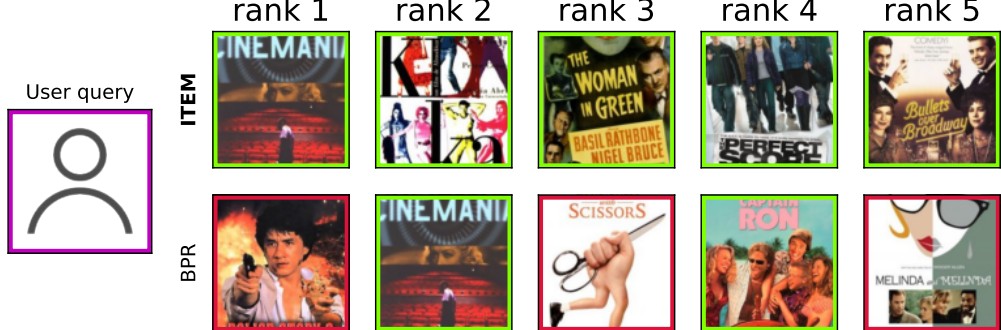

Figure 5: Qualitative results on MovieLens-100k. We compare the ranking obtained using LightGCN trained with the baseline BPR loss (bottom row) to the ranking obtained using **ITEM** (top row). Positive elements are highlighted in green, while negative elements are indicated in red.

strategy to sample negatives, relying on the Personalized PageRank score. It leverages the graph structure of a local neighborhood around a user: an item with a high PPR score for a user will be more challenging to distinguish for an MP-GNN, and enhance the learning performance.

Furthermore, we expanded the evaluation of GNNs by benchmarking on an inductive user-split protocol, which gauges the generalization capacity of GNNs. In both inductive and transductive protocols, we showcased the effectiveness of our ITEM learning framework, exhibiting improvements over the conventional BPR loss function, even compared to standard GNN training-enhancements methods such as SGL-ED, SIMGCL, or MixGCF. Our findings demonstrate that using ranking losses to enhance the training of MP-GNNs is very a promising research direction.

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

## A Method

In this section we define several ranking-based metrics used to evaluate recommender systems in Appendix A.1. In Appendix A.2 we show how to use the rank approximation of Eq. (1) of main paper to define other differentiable ranking-based losses.

### A.1 Metric definition

We remind here the definition of other well-known ranking-based metrics, used to evaluate recommender systems:

$$R@k = \frac{\# \text{ of positive in the top-}k}{\min(|\mathcal{V}^+|, k)} = \frac{1}{\min(|\mathcal{V}^+|, k)} \sum_{p \in \mathcal{V}^+} H(k - \text{rank}(p)) \tag{5}$$

$$\text{AP} = \frac{1}{|\mathcal{V}^+|} \sum_{p \in \mathcal{V}^+} \frac{\text{rank}^+(p)}{\text{rank}(p)}, \text{ with } \text{rank}^+(p) = \sum_{j \in \mathcal{V}^+} H(s_j - s_p) \tag{6}$$

### A.2 Other ranking based losses

In this section using the differentiable rank approximation of Eq. (1) of the main paper we define other differentiable ranking-based losses. In Appendix B.4 we show the choice of the metric used during training impacts the evaluation metric.

The differentiable Average Precision loss is defined as follows (Qin et al. (2009); Brown et al. (2020)):

$$\mathcal{L}_{\text{ITEM}}^{\text{AP}} = 1 - \frac{1}{|\mathcal{V}^+|} \sum_{p \in \mathcal{V}^+} \frac{\text{rank}_s^+(p)}{\text{rank}_s(p)}, \text{ with } \text{rank}_s^+(p, \tau) = 1 + \sum_{j \in \mathcal{V}^+} \sigma(s_j - s_p, \tau) \tag{7}$$

Note that we use the approximation of Eq. (1) of the main paper to approximate both rank and rank$^+$, using the same temperature $\tau$ parameter.

Finally, Patel et al. (2022) defines an approximation the R@k as follows:

$$\mathcal{L}_{\text{ITEM}}^{R@k} = 1 - \frac{1}{|\mathcal{K}|} \sum_{k \in \mathcal{K}} \frac{1}{\min(|\mathcal{V}^+|, k)} \sum_{p \in \mathcal{V}^+} \sigma(k - \text{rank}_s(p), \tau^*) \tag{8}$$

In (Patel et al. (2022)) the loss uses different level of recalls, *i.e.* for $k$ in $\mathcal{K}$, it is necessary to provide enough gradient signal to all positive items. To train $\mathcal{L}_{\text{ITEM}}^{R@k}$, it is also necessary to approximate a second time the Heaviside function, using a sigmoid with temperature factor $\tau^*$. Note that one other interesting property of both $\mathcal{L}_{\text{ITEM}}$ and $\mathcal{L}_{\text{ITEM}}^{\text{AP}}$ is that they only require to select one hyper-parameter, $\tau$.

## B Experiments

### B.1 Datasets

We evaluate our approach in the implicit feedback recommendation task Zhang et al. (2010), simulating real-world scenarios where explicit feedback is too costly. The relationships are binary: if a user interacts with an item, it indicates their appreciation, represented by a link in the graph. We validate our method on four recommendation datasets. We use the latest small MovieLens-100k dataset (Harper & Konstan (2015)) in the inductive settings. It features relations between users and movies, where users give ratings to different movies. For implicit feedback, we keep relations with ratings of 3 or higher. We use the 2018 edition of the dataset from the Yelp challenge (Asghar (2016)). Rated items are bars and restaurants, we follow (Wang et al. (2019); He et al. (2020)) to get implicit feedback. We also evaluate ITEM on the Amazon-Book dataset from Amazon-review, we use the same pre-processing as NGCF (Wang et al. (2019)) and LightGCN (He et al. (2020)). For the three datasets we apply a pre-processing to make sure that we have the same items for the training and for the evaluation. We follow Wang et al. (2019) and only keep users that have at least 10 interactions. Tab. 1 in main paper presents the datasets statistics.

### B.2 Implementation details

All our models are trained using the Adam optimizer, with learning rate in {0.01, 0.001}. On the inductive setting, we use $\tau = 1.0$, and $\tau = 1.5$ on the transductive setting. We use sum-pooling and embeddings of dimension 200 for the inductive setting on Yelp and MovieLens-100k and embeddings of dimension 64 for Amazon-Book, sum-pooling – except for LightGCN (He et al. (2020)) with mean-pooling – and embeddings of dimension 64 for the transductive setting. On both settings, we sample 5 positives and 200 negatives for ITEM for Yelp2018, MovieLens-100k and MovieLens-1M and 600 negatives for AmazonBook. We use, on both protocols, batch sizes of 512 for MovieLens-100k, MovieLens-1M, and 2048 for Yelp-2018 and Amazon-book. For the inductive setting, we found experimentally that learning user embeddings during training is harmful for the generalization of the GNNs to new users, so for training and evaluation users' embeddings are inferred using message passing only. For GIN (Xu et al. (2018)), GAT (Veličković et al. (2017)) and GCN (Kipf & Welling (2016)) we use the framework Pytorch Geometric (Fey & Lenssen (2019)) to implement the models.

### B.3 Ablation studies

Table 7: Comparison of our ranking-based loss ITEM vs the BPR loss (Rendle et al. (2012)), using different GNN architecture on 3 *inductive* benchmarks. For each architecture best results is **bold**, best overall results underlined.

| Method | | MovieLens-100k | | Yelp-2018 | | Amazon-book | |
|---|---|---|---|---|---|---|---|
| | | R@20 | NDCG@20 | R@20 | NDCG@20 | R@20 | NDCG@20 |
| **GCN** | BPR | 28.74 | 27.68 | 7.34 | 5.76 | 8.85 | 7.61 |
| | $\mathcal{L}_{\text{ITEM}}$ | **32.24** | **31.44** | **10.06** | **8.35** | **10.23** | **8.82** |
| | %Improv. | +12.2% | +13.5% | +37.1% | +45% | +15.6% | +15.9% |
| **GIN** | BPR | 29.71 | 27.58 | 8.25 | 6.74 | 9.62 | 8.05 |
| | $\mathcal{L}_{\text{ITEM}}$ | **32.00** | **29.87** | **10.57** | **8.79** | **13.05** | **11.59** |
| | %Improv. | +7.7% | +8.3% | +28.5% | +30.4% | +35.7% | +44.0% |
| **GAT** | BPR | 31.01 | 28.92 | 9.04 | 7.32 | 9.88 | 8.17 |
| | $\mathcal{L}_{\text{ITEM}}$ | **32.04** | **30.54** | **11.43** | **9.58** | **11.70** | **10.00** |
| | %Improv. | +3.3% | +5.6% | +26.4% | +30.9% | +18.4% | +22.4% |
| **LGCN** | BPR | 30.79 | 29.73 | 7.88 | 6.34 | 9.56 | 8.02 |
| | $\mathcal{L}_{\text{ITEM}}$ | **33.13** | **32.07** | **9.88** | **8.32** | **10.66** | **9.52** |
| | %Improv. | +7.6% | +7.9% | +25.4% | +31.2% | +11.5% | +18.7% |

#### B.3.1 Inductive loss comparison

In Table 7 we compare in the same settings the BPR loss (Rendle et al. (2012)) and our proposed ranking-based loss ITEM. We show that on the three inductive benchmarks, and across all four considered architectures, ITEM outperforms the BPR loss. On MovieLens-100k and with the best performing architecture, LightGCN (He et al. (2020)), our loss outperforms the BPR loss with +2.34 R@20 and +2.34 NDCG@20. The relative improvements are also significant, ranging from +3.3% NDCG@20 for GCN, up to 12.2% NDCG@20 for GCN. On Yelp-2018 which is a large scale dataset, GAT is the best performing architecture. ITEM outperforms the BPR loss by +2.39 R@20 and +2.26 NDCG@20. We can point out the considerable relative improvements on Yelp-2018, which are always larger than 25% and reaching 45% over GCN for NDCG@20. Finally on Amazon-book the best overall results are obtained with GIN using ITEM. It outperforms BPR and GIN by +3.4 R@20 and 3.5 NDCG@20 which are huge relative improvements on Amazon-book our most large scale dataset. Note that accros the three

considered datasets different GNN architectures work best, LightGCN (He et al. (2020)) on MovieLens-100k, GAT (Veličković et al. (2017)) on Yelp-2018 and GIN (Xu et al. (2018)) on Amazon-book. Overall, Table 7 – similarly as Tab. 4 in the main paper – shows the interest of optimizing a ranking-based loss, $\mathcal{L}_{\text{ITEM}}$, rather than the BPR loss, *i.e.* a *proxy* loss.

### B.3.2 Negative sampling

In Table 8 we show the impact of our PPR negative sampling strategy in the inductive setting. We show that on the three inductive benchmarks using our sampling strategy boost the performances of our trained GNN model over the random negative sampling (RNS), *e.g.* +0.66 R@20 and +0.4 NDCG@20 on Yelp-2018.

Table 8: Comparison of Random Negative Sampling (RNS) vs the sampling method in ITEM (PPR). We use a LightGCN (He et al. (2020)) backbone for the three **inductive** benchmarks.

| Loss | Sampling | MovieLens-100k | | Yelp-2018 | | Amazon-book | |
|------|----------|-----|---------|-----|---------|-----|---------|
| | | R@20 | NDCG@20 | R@20 | NDCG@20 | R@20 | NDCG@20 |
| $\mathcal{L}_{\text{ITEM}}$ | RNS | 33.13 | 32.07 | 9.88 | 8.32 | 10.66 | 9.52 |
| | PPR (ours) | **33.84** | **32.63** | **10.54** | **8.70** | **11.03** | **9.89** |

### B.3.3 Cold-start evaluation

In Table 9, we evaluate our model in a cold-start setting with only a small proportion of training data available for each user (10%, 20%, 50%). Our method, which uses an approximation of NDCG and PPR-based negative sampling, shows significant robustness compared to the baseline LightGCN with BPR loss.

Table 9: Cold-start evaluation in a transductive settings on MovieLens-1M with different amount of training data (%).

| Method | 10% | | 20% | | 50% | |
|--------|-----|---------|-----|---------|-----|---------|
| | R@20 | NDCG@20 | R@20 | NDCG@20 | R@20 | NDCG@20 |
| LGCN-BPR | 15.88 | 15.43 | 20.05 | 19.18 | 24.19 | 23.36 |
| ITEM | **17.11** | **16.89** | **22.37** | **21.68** | **29.03** | **28.20** |

### B.4 Model analysis

### B.4.1 Metric optimization

In Table 10, we compare the benefits of optimizing different ranking-based losses on MovieLens-100k (inductive) using a LightGCN model. Specifically, we compare the optimization of AP, R@k and NDCG, and include the BPR loss as a baseline. First we can note that on each metric all our ranking-based losses outperform the BPR loss. We can see that, for each metric, using its smooth approximation to optimize a GNN during training yields a higher score on this target metric. $\mathcal{L}_{\text{AP}_s}$ yields the best score of 19.05 AP, $\mathcal{L}_{\text{R@}k_s}$ yields the best score of 33.37 R@20 (outperforming the results reported in Table 7). Finally $\mathcal{L}_{\text{NDCG}_s}$ yields the best score for both NDCG and NDCG@20 of 53.55 and 32.07 respectively.

### B.5 Qualitative Results

We display on Fig. 6 the top-20 items retrieved when using ITEM on MovieLens-100k, and on Fig. 7 the top-20 items retrieved when using the baseline BPR loss (Rendle et al. (2012)). Both models are LightGCN (He et al. (2020)). We can observe qualitatively that ITEM brings more positive results, and leads to a better ranking than the BPR loss.

Table 10: Performances of different ranking-based losses on the MovieLens-100k inductive benchmark, and the BPR loss Rendle et al. (2012) baseline. The model used is LightGCN He et al. (2020).

| Loss | AP | NDCG | R@20 | NDCG@20 |
|---|---|---|---|---|
| BPR | 16.56 | 50.68 | 29.59 | 28.24 |
| $\mathcal{L}_{\text{ITEM}}^{\text{AP}}$ | **19.05** | 53.48 | 32.61 | 31.91 |
| $\mathcal{L}_{\text{ITEM}}^{R@k}$ | 18.57 | 52.93 | **33.37** | 31.82 |
| $\mathcal{L}_{\text{ITEM}}$ | 18.94 | **53.55** | 33.13 | **32.07** |

Figure 6: Top-20 results on MovieLens-100k (inductive) when using LightGCN and **ITEM**.

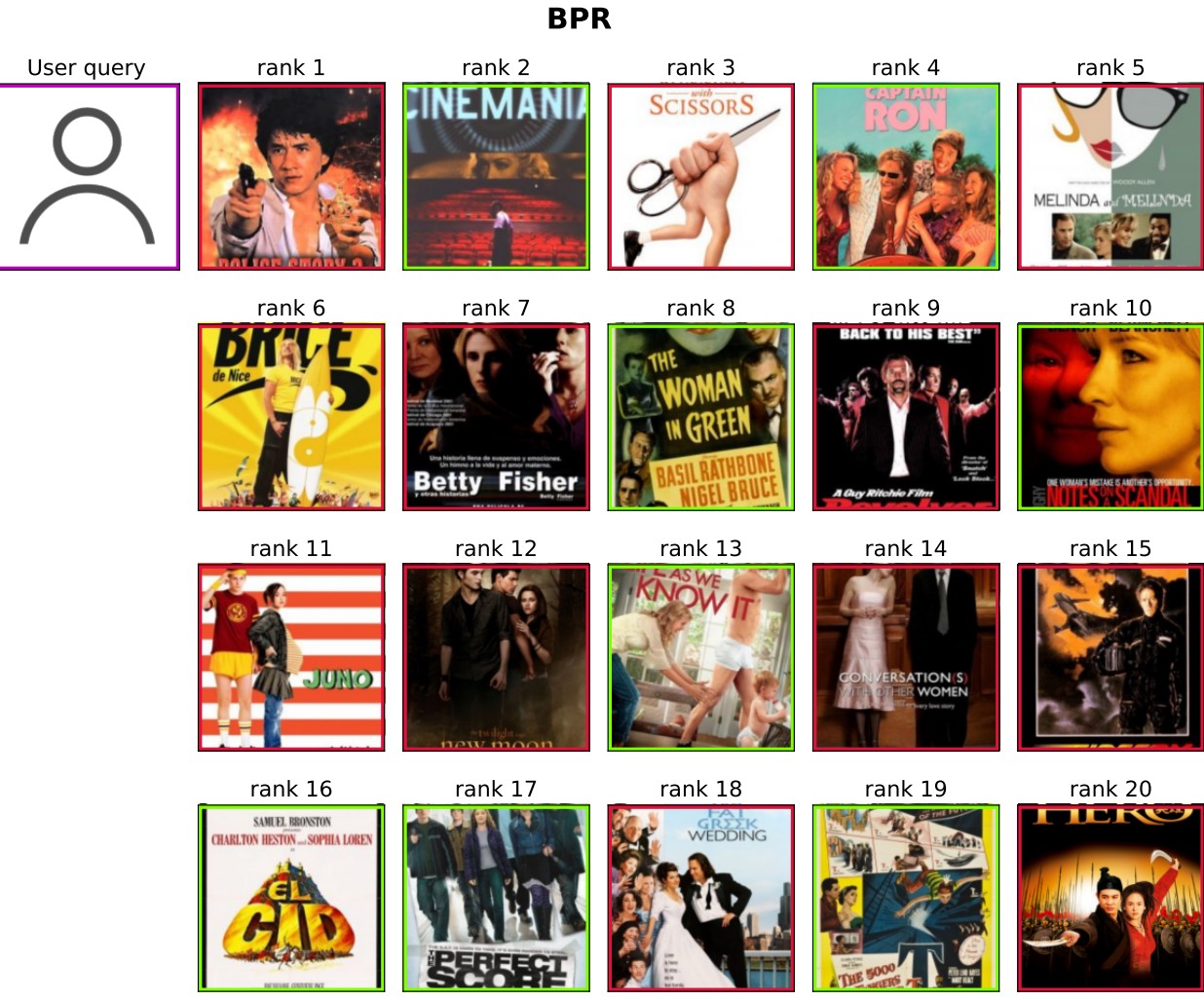

Figure 7: Top-20 results on MovieLens-100k (inductive) when using LightGCN and BPR loss.

