# OpenReview forum: "ITEM: Improving Training and Evaluation of Message-Passing based GNNs for top-k recommendation"
_TMLR — Accepted by TMLR_

### Review · Reviewer_jWmJ · 2024-02-23

**Summary Of Contributions:**

This paper addresses the problem of multiple recommendations with implicit feedback. It improves the learning phase in two ways. First, it uses a smooth approximation of the object rank computation to optimize a cost function that directly approximates the target ranking metrics. Secondly, the paper uses Personalized PageRank (PPR) to randomly draw negative examples that are as informative as possible. The corresponding approach is extensively evaluated experimentally, both in a transductive context (the training base contains part of the returns for each user), and in an inductive context (some users are not part of the training base).

**Audience:**

Yes

**Claims And Evidence:**

Yes

**Requested Changes:**

In Section 3, paragraph "Training Context", I guess the embedding $h$ is for any node in $\cal{U} \cup \cal{V}$ instead of $\cal{U} \times \cal{V}$.

**Strengths And Weaknesses:**

### Strengths
* Multiple recommendation with implicit feedback is a relevant problem for many applications.
* Rank function approximation enables direct optimization of the target utility function, and works over a wide range of temperature values.
* Experiments are very advanced.

### Notes
* The effect of selecting negative examples through PPR could be both beneficial and detrimental. On the one hand, we're highlighting items that are close to the decision frontier, and which therefore help the learning process. But, on the other hand, it forces the learned model to rank these items as negatively as possible, even though they are potentially the most relevant items for the user among those he hasn't considered so far. This aspect is all the more important because, at the same time, items that are much further away from the user are rarely selected by PPR, so the learned model is less encouraged to reject them.
However, experiments show that the beneficial aspect prevails, and the fact that negative examples are randomly drawn means that distant examples can be selected from time to time.

---

### Review · Reviewer_2cdS · 2024-02-26

**Summary Of Contributions:**

This paper studies the problem of recommender systems with graph neural networks. Specifically, the authors consider that the evaluation metrics can be directly optimized as loss functions in graph neural networks. The authors conduct experiments on four datasets, showing the best performance of the proposed method.

**Audience:**

No

**Broader Impact Concerns:**

There is no concern on the ethical implications of the work.

**Claims And Evidence:**

No

**Requested Changes:**

- The motivation is not enough. Optimizing the loss function directly is not a good choice in real-world recommender systems since metrics are too many. Furthermore, there are more metrics than NDCG. What about the performance when using other metrics?
- The baselines are out-of-date. There is no baseline published in 2023. As we know, the recommendation models are developing very fast.
- The method is relatively straightforward. There is no tailored solution for recommender systems, while it is only an adaption from general graph neural network-based approaches.

**Strengths And Weaknesses:**

- Recommender systems, as one of the most important applications of machine learning, well matches the scope of TMLR.
- The authors consider four widely-used datasets for evaluation.
- The proposed method achieves the best performance.

---

### Review · Reviewer_iMQg · 2024-05-14

**Summary Of Contributions:**

The authors made the following major contributions in this paper:
1. They proposed a ranking loss function to directly optimize NDCG in learning MP-GNN for recommendation while most existing work use a proxy (e.g. BPR).
2. They proposed to use Personalized PageRank for negative sampling, which is computationally more efficient and generate negative samples of higher quality, which eventually boosted the performance of their model.
3. They extended evaluation by benchmarking on an inductive user-split protocol, which better gauges the generalization ability of the model.
4. They conducted extensive experiments and compared with multiple existing baselines to prove the effectiveness of their framework. They also conducted thoughtful ablation experiments to understand the impact of each of their proposed ideas.

**Audience:**

Yes

**Claims And Evidence:**

Yes

**Requested Changes:**

This work is solid and it just needs some minor changes to get even better. I suggested the author considering making the following changes
1. Add some discussion of similarity/connections and differences v.s. BPR and other exiting ranking-based loss function.
2. Add more details of Personalized PageRank for negative sampling for replication purpose. There is also one interesting thing worth discussing, as we all know, negative samples are not strictly negative as users may not be exposed to them yet. Does PPR increase the risks of picking items users interested to but not exposed to yet? Or does it decrease the risk? Why?
3. Add more details of the used dataset. What type of user feedback does each of them have? 0/1 or ratings? If feedback are ratings, how to transform them to 0/1 feedback?
4. Add some evaluation & analysis for cold-start users if possible?
5. Correct the citation error at the end of Paragrah 1, Page 8.

**Strengths And Weaknesses:**

Strength:
1. The idea of using list-wise ranking loss function to directly optimize the evaluation metric makes a lot of sense. The authors successfully proposed a loss function and proved its usefulness.
2. The authors conducted extensive experiments and analysis to demonstrate the effectiveness of their framework, which makes this paper  empirically sound.
3. The authors adapted inductive user split protocol in their evaluation, which is important but has been overlooked a lot.

Weakness:
1. It would be great to have more discussions of the connection and comparison of the proposed idea v.s. existing work. For example, BPR is used as an important baseline in this work. In some sense, the proposed idea is similar to a transformation of BPR. What are the connections/similarities between them, what makes the proposed ideas more effective than BPR? Meanwhile, I did not see much discussions of existing ranking-based loss function and their differences to the proposed one. These discussion can help readers better understand the landscape of related research and better appreciate the proposed loss function.
2. It needs some justification about why the authors only consider 0/1 feedback in their loss function. In some cases, e.g. product/movie recommendation, the users may provide rating feedback other than 0/1.
3. The inductive user-split protocol is a great extension to evaluation. However, it does not cover the cold-start cases, where a user only have very few items. However does the framework work for cold-start users?

---

> ### Author Response · Authors · 2024-05-28
> **Response**
>
> We would like to thank you for your interesting questions, which we have tried to answer as best we can:
>
> 1. BPR is a pairwise loss function where, for each user, one positive and one negative item are sampled to optimize the model. This approach results in a coarse approximation of ranking, diverging from the actual task, which involves ranking a list of items from positive to negative for each user.  Our loss function, in contrast, is listwise, designed to directly optimize metrics like NDCG (see Section 3.1). It applies stronger penalties for positive-negative inversions at the top of the list compared to the bottom, ensuring better alignment between training and evaluation tasks. This alignment explains the superior performance of our loss function compared to BPR (see Table 4).
> Additionally, we use the sigmoid function to approximate NDCG, resulting in better performance. While approaches like ROADMAP [1] proposed other approximations of ranking, these did not yield satisfactory results in practice for the implicit feedback recommendation task.
> We provide a more detailed discussion on state-of-the-art ranking-based loss functions in Section 2.3, with details available in Appendix A2.
>
> 2. We calculate PPR scores for each user, evaluating their proximity to every item in the bipartite graph. These scores are normalized with the softmax function to create a probability distribution for sampling, as detailed in Section 3.2.As you rightly point out, our PPR-based Negative sampling strategy may select items that users are interested in but haven’t yet seen, increasing the risk of false negatives. However, this tradeoff is intentional. By selecting “hard negatives,” we improve the quality of the negative samples, enhancing the learning process. Table 5 shows that our method (ITEM-PPR) outperforms random negative sampling (ITEM-RNS). These results highlight that having more informative negative samples, even with some false negatives, leads to better model performance because hard negatives provide more challenging and useful learning signals.  It is better to work with a few false negatives than to use too many non-informative negatives.
> We included more detailed information about our PPR sampling strategy in Section 3.2.
>
> 3. Using scores of 0 or 1 aligns with the classic implicit feedback recommendation task [2]. This approach is common in recommendation systems and is the context for our work [3,4,5].
> In a real-world scenario, it corresponds to the situation in which a user clicks on a product or a video in an online catalog, providing insight into their preferences without explicit ratings. For datasets with non-binary feedback, a threshold may be applied to binarize them. For the MovieLens datasets, we consider movies rated 3 or higher as having a score of 1, while those rated below 3 have a score of 0. For the Yelp2018 and Amazon-Book datasets, we used the implicit-feedback transformation introduced by [4], where ratings above a certain threshold are transformed into a link in the graph. Also, in [4] they keep only users that have at least 10 interactions, and we follow this preprocessing by using their transformed data. However, it would be a great perspective for future work to experiment  applying these rank optimization functions to data where ratings are available (explicit feedback), as the NDCG metric also measures the quality of a ranking based on relevance.
> We have added those details to the datasets descriptions in Appendix B.1.
>
> 4.   Add cold-start eval : We conducted an evaluation of our model in a cold-start setting with a limited amount of training data for each user (10%, 20%, 50%). The results in Table 9 of Appendix B.3.3 show that our method, which uses an approximation of NDCG and PPR-based negative sampling, significantly outperforms the baseline LightGCN with BPR loss.
> Our approach consistently achieves higher Recall@20 and NDCG@20 scores, demonstrating its robustness and effectiveness in cold-start scenarios. This confirms that our method provides better performance by using more informative negative samples, even with the trade-off of potential false negatives, compared to random non-informative negatives.
> The table  9 of Appendix B.3.3 was added to the article to respond to your comment.
>
> 5. Error in a citation : We thank you for pointing out this citation error, which we have corrected in our updated article.
>
>
> [1] Ramzi, E., et. al. Robust and Decomposable Average Precision for Image Retrieval. Neurips 2021
> [2] Bangzuo Zhang,et al.  “Survey of user behaviors as implicit feedback,” 2010 International Conference on Computer, Mechatronics, Control and Electronic Engineering, Changchun
> [3] He, X et al. LightGCN: Simplifying and Powering Graph Convolution Network for Recommendation. SIGIR 2020
> [4] Wang, X.et al. (2019). Neural Graph Collaborative Filtering. SIGIR 2019
> [5] Liang, D. et al. Variational autoencoders for collaborative filtering. WWW-2018

---

### Author Response · Authors · 2024-05-28
**Updated article**

Dear editor and reviewers,

We would like to warmly thank you and the reviewers for the constructive feedback on our manuscript. We have carefully considered all comments and have made corresponding revisions to our manuscript. We think that the suggestions the reviewers have asked for helped us in significantly improving the paper. New parts and modifications to the article are written in blue. Specifically:

- We extended state-of-the-art with a discussion on rank-based losses.

- We have clarified and strengthen our explanation of the PPR sampling strategy.

- We have included more baselines for some experiments and a few more experiments, to improve the robustness of our results.

We hope that the revisions we have made address the reviewers' concerns satisfactorily. We truly appreciate the time and effort that you and the reviewers have invested in reviewing our manuscript and look forward to your feedback.

Sincerely,

---

### Decision · Action_Editor_ePZu · 2024-06-30

**Recommendation:** Accept as is

**Comment:**

The comments by the reviewers were generally positive, reporting negative aspects mainly concerning presentation, motivation, and some additional justification. One reviewer asked for more experimental evidence involving more recent approaches, which authors provided, although the reviewer seems not to become aware of this addition to the experimental assessment. In the updated version, authors also responded in a satisfactory way to the comments by the other two reviewers. Overall, the contribution of the paper is relevant, timely, and solid.

**Audience:**

TMLR's audience will be interested to the topic covered by the paper, since recommendation systems are an important area, where ML techniques are broadly applied.

**Claims And Evidence:**

The claim made in the paper are supported by convincing evidence. In the updated version of the paper, after the discussion with reviewers, an experimental comparison versus a more recent approach confirmed the original claims.